# Improved Split TEV GPCR β-arrestin-2 Recruitment Assays via Systematic Analysis of Signal Peptide and β-arrestin Binding Motif Variants

**DOI:** 10.3390/bios13010048

**Published:** 2022-12-29

**Authors:** Yuxin Wu, Isabelle V. von Hauff, Niels Jensen, Moritz J. Rossner, Michael C. Wehr

**Affiliations:** 1Research Group Cell Signalling, Department of Psychiatry and Psychotherapy, University Hospital, Ludwig Maximilian University of Munich, Nussbaumstr. 7, 80336 Munich, Germany; 2Section of Molecular Neurobiology, Department of Psychiatry and Psychotherapy, University Hospital, Ludwig Maximilian University of Munich, Nussbaumstr. 7, 80336 Munich, Germany; 3Systasy Bioscience GmbH, Balanstr. 6, 81699 Munich, Germany

**Keywords:** GPCR, drug screening, drug discovery, cell-based assay, split TEV technique

## Abstract

G protein-coupled receptors (GPCRs) are major disease-relevant drug targets; robust monitoring of their activities upon drug treatment is key to drug discovery. The split TEV cell-based assay technique monitors the interaction of an activated GPCR with β-arrestin-2 through TEV protein fragment complementation using a luminescent signal as the readout. In this work, split TEV GPCR β-arrestin-2 recruitment assays were optimized to monitor the endogenous ligand-induced activities of six GPCRs (DRD1, DRD2, HTR2A, GCGR, AVPR2, and GLP1R). Each GPCR was tested in four forms; i.e., its wildtype form, a variant with a signal peptide (SP) to facilitate receptor expression, a variant containing the C-terminal tail from the V2 vasopressin receptor (V2R tail) to promote β-arrestin-2 recruitment, and a variant containing both the SP and V2R tail. These 24 GPCR variants were systematically tested for assay performance in four cell lines (HEK-293, PC12 Tet-Off, U-2 OS, and HeLa). We found that the assay performance differed significantly for each GPCR variant and was dependent on the cell line. We found that V2R improved the DRD2 split TEV assays and that HEK-293 cells were the preferred cell line across the GPCRs tested. When taking these considerations into account, the defined selection of assay modifications and conditions may improve the performance of drug development campaigns that apply the split TEV technique as a screening tool.

## 1. Introduction

G protein-coupled receptors (GPCRs), which constitute the largest class of cell surface receptors, regulate various biological processes in health and disease, including the proliferation and differentiation of cells, neuronal activity, immune response, hormonal modulation, vision, taste, and smell [1,2]. Abnormal GPCR activity and altered downstream cellular signaling is implicated in various human diseases such as cancer and diabetes as well as in neurological and psychiatric disorders, including schizophrenia [3,4]. Because GPCRs have critical roles in the pathophysiology of many complex diseases, they are key drug targets. Consequently, GPCRs are currently targeted by 33% of all marketed drugs, which makes them the largest druggable class of receptors [5].

GPCRs are seven-transmembrane receptors that have three intracellular loops and a C-terminal tail for transducing cellular signals [1]. G proteins and β-arrestins can bind to those intracellular regions of GPCRs, thereby constituting options for biased signaling [6,7]. The previously canonical signaling route is via G proteins that can be categorized into stimulating or inhibitory effectors that initiate specific downstream signaling events such as cAMP-mediated or calcium-dependent pathways [8]. Activation of GPCRs causes the G protein-dependent phosphorylation of the C-terminal tail by G protein-coupled receptor kinases (GRKs), which provides docking sites for β-arrestins. Arrestin recruitment to the GPCR leads to the desensitization of the primary signaling response by internalizing the GPCR/β-arrestin complex and initiates cellular signaling responses that include the activation of ERK signaling. Importantly, G protein- vs. β-arrestin-mediated signaling, also known as biased signaling, defines the physiological response of an activated GPCR [8,9]. While, for example, Gαs proteins link GPCR activity to increased cAMP signaling, activated β-arrestins initiate mitogen-activated protein kinase (MAPK) signaling cascades. In addition, by deciphering these mechanisms of cellular signaling, there is the opportunity to create better drugs with fewer side effects [10].

In drug discovery, cell-based assays are often applied for GPCR-targeting drugs [11]. For example, the activity of GPCRs can be monitored via genetically encoded reporter gene assays: either indirectly using pathway assays such as CREB responsive element (CRE) assays that depend on cAMP and calcium levels [12] or directly using target-based assays that are based on β-arrestin-2 recruitment and rely on proximity [13,14]. One type of such a GPCR/β-arrestin-2 recruitment assay is based on the split TEV assay, which is based on the functional complementation of TEV protease fragments [15,16]. When brought into proximity, the TEV protease fragments functionally complement to form an active protease and release a GPCR-anchored transcriptional co-activator that migrates into the nucleus to initiate a reporter gene of choice such as firefly luciferase. Whereas CRE pathway assays use the physiology of the cell and depend on intracellular cAMP levels, target-based assays such as the split TEV assay enable the direct monitoring of GPCR activity. The C-terminal intracellular domain of the vasopressin receptor 2 (V2R tail, amino acids 343–371) contains 11 serine and threonine residues that can be phosphorylated by GRKs and constitutes a docking site for β-arrestin-2. The V2R tail has been shown to promote assay performance for full-length TEV GPCR/β-arrestin-2 recruitment assays [17]. Therefore, the V2R tail has also been applied to split TEV GPCR assays [15,16]. Furthermore, the fusion of a cleavable signal peptide (SP) derived from influenza virus hemagglutinin to the N-terminal end of a GPCR was reported to enhance surface expression and to be beneficial for assay performance [18].

Here, we describe a systematic approach to develop sensitive and robust target-based split TEV assays for GPCRs in which we measured whether an additional artificial SP and/or the V2R tail fused to a GPCR improved assay performance. We applied this approach to six GPCRs of various subfamilies, including class A receptors (the dopamine receptors DRD1 and DRD2, serotonin receptor 2A (HTR2A), and vasopressin receptor 2 (AVPR2)) and class B receptors (the glucagon receptor (GCGR) and glucagon-like peptide 1 receptor (GLP1R)). In addition, we tested assay performances in various cell lines used for GPCR biology, including HEK-293, U2-OS, HeLa, and PC12 Tet-Off (PC12-TO) cells. We found that it was critical to test whether the addition of the artificial SP and/or the V2R tail helped to establish a sensitive and robust assay in a given cell line because in most cases, the addition of either the SP or V2R tail compromised the assay performance as measured by the fold change and the Z’ factor [19]. DRD2 assays benefited from the addition of the V2R tail to improve the assay performance in HEK-293 cells. In contrast, for the other GPCRs tested, the fusion of the SP had a negligible effect on the assay performance. In addition, we found that the cell line of choice was critical for a given split TEV GPCR assay.

## 2. Materials and Methods

### 2.1. Plasmids

The GPCR ORFs were amplified via PCR using the Q5 High-Fidelity DNA Polymerase (NEB), and the resulting PCR was BP-recombined into the pDONR/Zeo plasmid using Gateway recombination cloning (Thermo Fisher Scientific, Waltham, MA, USA). Each entry clone plasmid was control-digested using BsrGI, which cut inside the recombination sequences and thus released the insert. Lastly, the GPCR ORF sequences were verified via Sanger sequencing. Gateway LR recombination was used to transfer the ORFs from the entry vectors into the split TEV destination vectors (either pcDNA3_attR1-ORF-attR2-NTEV-TCS-GV-2xHA_DEST or pcDNA3_attR1-ORF-attR2-V2R-NTEV-TCS-GV-2xHA_DEST). The signal peptide (SP; peptide sequence: MKTIIALSYIFCLVFA↓DYKDDDDASID, cleavage site indicated by the arrow) derived from hemagglutinin [13] was added via 2-step PCR to the GPCR ORFs. Gateway entry clones for DRD2, HTR2A, and AVPR2 without SP, as well as the Gateway expression clone for ARBB2-CTEV (pcDNA3.1_Zeo_ARRB2-1-383-CTEV-2xHA), were described previously [16]. The Gateway entry clone for GLP1R was obtained from Harvard PlasmID (pENTR223-1_GLP1R_Cop, HsCD00082670). The plasmids used in this study are listed in Appendix A and are available at Addgene. The oligos used for cloning are listed in Appendix A. 

### 2.2. Compounds

The dopamine hydrochloride, [Arg8]-vasopressin acetate salt, and glucagon were purchased from Sigma-Aldrich (St. Louis, MO, USA). The serotonin hydrochloride was obtained from Tocris. The liraglutide (NN2211) was purchased from Selleck Chemicals (Houston, TX, USA).

### 2.3. Cell Culture

The HEK-293 (ATCC, CRL-1573) and HeLa (ATCC, CCL-2) cells were cultured in DMEM (4.5 g/L glucose, Thermo Fisher Scientific) supplemented with 2 mM GlutaMAX (Thermo Fisher Scientific), 10% FCS (Thermo Fisher Scientific), 100 U/mL of penicillin, and 100 µg/mL of streptomycin (Thermo Fisher Scientific). The PC12 Tet-Off cells (Clontech, 631134; PC12-TO) were maintained in DMEM medium (1 g/l glucose, Thermo Fisher Scientific) supplemented with 10% FCS, 5% horse serum (HS), 2 mM GlutaMAX, 100 U/mL penicillin, and 100 µg/mL streptomycin (all Thermo Fisher Scientific). The osteosarcoma U-2 OS cells (ATCC, HTB-96) were cultured in McCoy’s 5A (Modified) Medium supplemented with GlutaMAX containing 10% FCS, 100 U/mL penicillin, and 100 µg/mL streptomycin (all Thermo Fisher Scientific). The PC12-TO cells were grown on surfaces coated with poly-l-lysine (PLL, Sigma-Aldrich) for the maintenance and experiments. For coating, the plates were incubated with PLL (0.02 mg/mL final concentration diluted in ddH_2_O) for 30 min at 37 °C, washed twice with ddH_2_O, and air-dried. The cells were cultured at 37 °C and 5% CO_2_.

### 2.4. Luciferase Assays

For the luciferase assays, the cells were plated on flat-bottom 96-well clear plates (Falcon) at 2 × 10^4^ HEK-293 cells/well, at 4 × 10^4^ HeLa cells/well, at 4 × 10^4^ U-2 OS cells/well, or at 5 × 10^4^ PC12-TO cells/well one day before the experiment. All of the luciferase assays were performed using 6 replicates per condition. For the split TEV assays, the cells were transfected with split TEV plasmids (GPCRs and β-arrestin-2) and the UAS reporter plasmid (pGL4_10xUAS-MLPmin-luc2). For the CRE pathway assays, the cells were transfected with GPCR and a CRE reporter (pGL4_CRE-CMVmin-luc2) plasmids. All of the transfection mixes also contained a plasmid that encoded a nuclear variant of EYFP (1 ng per 96-well plate) for visual control of the transfection efficiency. Thus, we consistently transfected 31 ng for the split TEV assays and 21 ng for the CRE assays per 96-well plate. The transfection was conducted according to the manufacturer’s instructions. The plasmids and the transfection reagent (Turbofect (Thermo Fisher Scientific) for the HEK-293 cells at a ratio of 1 µg of DNA to 3 µL of Turbofect; and Lipofectamine 3000 (Thermo Fisher Scientific) for the U2-OS, HeLa, and PC12-TO cells at a ratio of 1 µg of DNA to 2 µL of P3000 and 1 µg of DNA to 1.5 µL of Lipofectamine 3000) were diluted in Opti-MEM (Thermo Fisher Scientific) and incubated for 20 min at room temperature and added to the cells. Next, the medium was removed from the cells, the transfection mix was added, and the assay plates were incubated for 2 h at 37 °C. Double the volume of the culture medium (final volume: 90 µL per well) was then added to dilute the transfection reagents. The next day, the culture medium was replaced with serum-free assay medium (HEK-293: DMEM (4.5 g/L glucose) supplemented with 2 mM GlutaMAX; HeLa: DMEM (4.5 g/L glucose) supplemented with 2 mM GlutaMAX; U-2 OS: McCoy’s 5A medium; and PC12-TO: serum-reduced medium (DMEM (1 g/L glucose) supplemented with 1% dialyzed FBS (Thermo Fisher Scientific), 2 mM GlutaMAX, and 0.1 mM non-essential amino acids (NEAA, Thermo Fisher Scientific)) for 17–18 h. On the second day, the medium was removed, and the cells were treated with compounds diluted in the assay medium at various concentrations for 6 h at 37 °C. For the dose–response analyses, the compounds were diluted on a semi-logarithmic scale using 15 concentrations that ranged from 1 pM to 10 µM. Next, the medium was removed, and the cells were lysed with 30 µL of passive lysis buffer (Promega, Madison, WI, USA). To measure the firefly luciferase activity, the lysates were transferred to white flat-bottom 96-well plates (Falcon, Minato City, Tokyo). The firefly luciferase activity was measured with a Mithras LB 940 Microplate Reader (Berthold Technologies, Bad Wildbad, Germany) using the MicroWin 2000 software. The data were exported to Excel and processed with R-based scripts based on the *ggplot2* package to calculate and plot bar graphs with mean values, the standard deviation (s.d.), and the data points of the 6 replicates. To plot and analyze the dose–response curves, the R-based *drc* package [20] was used with the four-parameter log logistic function for curve fitting. The data were plotted as means with the standard error of the mean (s.e.m.) of the 6 replicates of the firefly readings. The assays were repeated twice.

### 2.5. Western Blotting and Antibodies

To measure the expression levels of the split TEV GPCR fusion constructs, the plasmids were transfected into HEK-293 cells using Lipofectamine 3000 according to the manufacturer’s instructions. After allowing the plasmids to express for 24 h, the cells were washed 1× with PBS and lysed in a Triton-X lysis buffer (1% Triton-X100, 50 mM Tris pH 7.5, 150 mM NaCl, and 1 mM EGTA) containing the Complete protease inhibitor cocktail (Roche, Basel, Switzerland) and the PhosSTOP phosphatase inhibitor (Roche). The lysed cells were kept on ice for 10 min, sonicated 3× for 10 s at 4 °C, and denatured for 10 min at 70 °C. The Mini-PROTEAN Tetra Electrophoresis System (Bio-Rad, Hercules, CA, USA) was used for running and blotting the protein gels. For the chemiluminescence detection of proteins, the Pierce ECL Western Blotting Substrate (Thermo Fisher Scientific) was used followed by imaging with a ChemoStar ECL imager (Intas Science Imaging Instruments, Göttingen, Germany). The HA-tagged proteins were visualized using an HA antibody (clone 3F10, dilution 1:500, No. 11 867 423 001, Roche).

### 2.6. Immunocytochemistry of Cells

A total of 200,000 HEK293 cells were seeded per 24 wells, and 200 ng of each GPCR plasmid was transfected using Lipofectamine 3000. The cells were allowed to express the plasmids for 24 h and then were washed with PBS and fixed with 4% paraformaldehyde for 10 min at room temperature. Next, after three washes with TBS for 5 min each, the cells were blocked with 3% BSA in TBS for 1 h at room temperature. To stain for surface expression of SP-GPCRs (note that the SP harbored a FLAG tag), the cells were not permeabilized, but rather directly incubated with an anti-FLAG M2 antibody (dilution 1:500, F1804, Sigma-Aldrich). For the intracellular staining (note that the HA tag was intracellular), the cells were permeabilized via two washes with TBS/Triton X-100 (0.1%) (TBS-T) for 5 min each, blocked with 3% BSA in TBS-T for 1 h at room temperature, and incubated with an anti-HA antibody (clone 3F10, dilution 1:500, No. 11 867 423 001, Roche). The primary antibodies were incubated for 3 h at room temperature. Next, the cells were washed 3× with TBS (or TBS-T for intracellular staining) and incubated for 1 h at room temperature with fluorescent conjugate cross-adsorbed secondary antibodies (Alexa 488 and Alexa 647, Thermo Fisher Scientific) at a dilution of 1:500. The cells were mounted in EverBrite™ Hardset Mounting Medium (Biotium, Fremont, CA, USA) that contained Dapi for nuclear staining. The cells were imaged on a ZEISS Axio Observer.Z1 microscope with a C-Apochromat 63×/1.20 W Corr objective.

## 3. Results

### 3.1. Construction of a Versatile Split TEV GPCR Assay Expression System Using Gateway Recombination Cloning

We previously established high-throughput applicable split TEV β-arrestin-2 recruitment assays for various GPCRs, including DRD1, DRD2, AVPR2, and HTR2A [15,16]. Given the importance of GCGR and GLP1R as therapeutic targets in type II diabetes [21,22] and obesity [23,24], as well as GLP1R’s implication in neurological disorders such as Alzheimer’s disease (AD), Parkinson’s disease (PD), and amyloid lateral sclerosis (ALS) [21,25,26], we established split TEV assays for those receptors as well. In our split TEV assay, the GPCR was fused to the N-terminal moiety of the TEV protease (NTEV), a TEV protease cleavage site (TCS), and the artificial transcriptional co-activator GAL4-VP16 (GV). The ligand-activated GPCR bound to β-arrestin-2, which, as a truncated version, was fused to the C-terminal moiety of the TEV protease (CTEV). Binding of the GPCR to β-arrestin-2 led to the functional complementation of the TEV protease fragments, thereby resulting in proteolytic activity. The TEV protease-cleaved GV migrated to the nucleus to bind to clustered upstream activated sequences (UAS) and initiated the transcription of a firefly luciferase reporter gene (Figure 1A). The GPCR in such a split TEV assay could be modified (1) with the N-terminally fused cleavable SP to enhance surface expression and (2) with a C-terminally fused V2R tail to enhance β-arrestin-2 binding to promote the assay performance (Figure 1B) [13,17]. The V2R tail may be particularly important for GPCRs with weak or absent endogenous activity-dependent β-arrestin-2-binding, which is also the case for DRD2 [27]. To test this, we selected six GPCRs from classes A and B that had a varying number of serine (Ser) and threonine (Thr) residues in their C-terminal tail. While DRD1, HTR2A, AVPR2, GCGR, and GLP1R all have 11 or more serine/threonine residues in combination with a different total length of the C-terminal tail, DRD2 had a very short C-terminus without Ser/Thr residues (Appendix A). To systematically test whether the V2R tail and SP improved the split TEV assay performance, we used the Gateway recombination system to clone for each GPCR (1) its native form, (2) a variant with an N-terminal SP (SP-GPCR), (3) a variant fused to the V2R tail (GPCR-V2R), and (4) a variant fused to both the SP and V2R (SP-GPCR-V2R) (Figure 1B). To confirm that each of the 24 GPCR constructs (Appendix A) was correctly expressed, we transfected all GPCR-NTEV-TCS-GV fusions into HEK-293 cells. The expression of each GPCR construct was validated via Western blotting against an HA tag that was present in all of the GPCR fusions (Figure 1C–E). The surface expression of GPCR split TEV fusions was validated via immunocytochemistry staining. While the localization for both the native and SP-GPCR-V2R variants was examined with an antibody against the C-terminal HA tag (Figure 1F,G), the SP-GPCR-V2R variants were additionally tested against an extracellular FLAG epitope located C-terminally to the SP (Figure 1H).

### 3.2. Performance of Split TEV GPCR Assays Depended on Modifications and Cell Type

Next, transient split TEV GPCR β-arrestin-2 recruitment assays were conducted for all 24 constructs in HEK-293 cells (Figure 2A–F), U2-OS cells (Appendix A), HeLa cells (Appendix A), and PC12-TO cells (Figure 3A–F). Each GPCR was stimulated with its cognate agonist for 6 h in accordance with previous continuous online luciferase assays for split TEV GPCR assays [16]. GLP1R was stimulated with the peptide mimetic agonist liraglutide [28]. For each assay, the fold-change ratios based on the firefly luciferase values of non-stimulated and stimulated samples were calculated (Table 1). We found that the assays for DRD1, DRD2, AVPR2, and GCGR performed best in the HEK-293 cells (Figure 2, Table 1), while the HTR2A and GLP1R assays performed best in the PC12-TO cells (Figure 3C,F, Table 1). Specifically, the highest fold changes were obtained with the native forms of DRD1, DRD2, AVPR2, and GCGR in the cells and the native forms of HTR2A and GLP1R in the PC12-TO cells. The Z′-factor is an indicator of assay performance that integrates both the baseline and activated means as well as the standard deviations thereof to determine a statistical effect size [19]. Excellent assays; i.e., assays with a large separation window that are compatible with high-throughput applications, have Z′-factor values between 0.5 and 1.0. Split TEV GPCR assays with the highest fold-change values correlated largely with the highest Z′-factors when considering only the optimal cell line for a top-performing assay (i.e., for DRD1, DRD2, AVPR2, and GCGR in HEK-293 cells) (Table 1). 

GPCRs fused to the SP and/or the V2R tail mostly retained their biological function with respect to mediating an agonist-induced β-arrestin-2 recruitment, but in some cases resulted in a considerable loss of activation in a cell-type-dependent manner; e.g., for HTR2A and GLP1R in the HeLa cells and for HTR2A in the U-2 OS cells (Table 1). The baseline luciferase readings of the SP-GPCR assays were invariably larger than the ones of the native forms, which also was reflected by the higher expression levels of the SP-GPCRs observed in the Western blot analysis (Figure 1C–E and Figure 2A–F).

For the DRD2 assays, the addition of the V2R tail considerably improved the assay performance in the HeLa and U-2 OS cells as determined by the Z′-factors (Appendix A, Table 1), while in HEK-293 cells, the additional V2R tail had a less pronounced effect (Figure 2B, Table 1). Nonetheless, the DRD2 assays overall performed best in the HEK-293 cells when taking both the Z′-factor and fold change into account. The addition of the V2R tail, especially when combined with the SP, to the DRD2 substantially increased the absolute luciferase reporter readings in the HEK-293, HeLa, and U-2 OS cell lines, but the overall performance was the best in the DRD2-V2R fusion. In contrast, the DRD2 assays did not perform in the PC12-TO cells at all. Of note, the GLP1R assays that used the native form and the V2R variant also performed robustly in the HEK-293 cells in terms of both the fold change and the Z′-factor. The preferred variant and cell line for each GPCR tested is summarized in Table 2.

### 3.3. Split TEV Assays for GCGR and GLP1R Correlated with a cAMP Response Element Pathway Assay

Here, we report for the first time the establishment of split TEV β-arrestin-2 assays for GCGR and GLP1R. To further characterize the assays for these two GPCRs in terms of sensitivity and robustness, we conducted dose–response assays with their respective ligands: glucagon (Figure 4A) and liraglutide (Figure 4B). HEK-293 cells were used for both assays because the GLP1R assays were robust both in HEK-293 cells (Figure 2F) and PC12-TO cells (Figure 3F). In the GCGR assay, glucagon had an EC_50_ of 4 nM, while liraglutide yielded an EC_50_ of 1 nM in the GLP1R assay. GCGR and GLP1R signaled via Gαs proteins and the second messenger cAMP to regulate the cellular signaling [21,29]. Therefore, the activity of these GPCRs could also be indirectly monitored using a cAMP response element (CRE) sensor assay [12]. To compare the target-based split TEV assays with this widely used cellular pathway assay, we performed dose–response assays in HEK-293 cells by transfecting the GCGR and GLP1R constructs with a CRE reporter plasmid that drove a firefly luciferase reporter gene. Both the glucagon (Figure 4C) and liraglutide (Figure 4D) treatments yielded dose-dependent responses: an EC_50_ of 25 nM for glucagon and of 7 nM for liraglutide, which were in the range of previously reported data obtained in HEK-293 cells [30].

## 4. Discussion

Arrestin-2-mediated transduction of GPCR signaling is nearly always assayed at the level of recruitment because downstream effectors or second messengers do not converge on a single pathway [8,10]. Split TEV GPCR assays are highly sensitive and directly transform GPCR-triggered β-arrestin-2 recruitment into luminescent signals. They are regularly used to measure GPCR activity in compound screens [15,16,31]. To establish split TEV assays with improved characteristics, we designed vectors compatible with Gateway recombination cloning to shuffle GPCRs of interest with and without an SP and V2R tail into expression cassettes to identify the most sensitive and robust assay. We tested six GPCRs that were selected from class A and class B subfamilies that were either stimulated by small molecules (biogenic amines) or peptide ligands to cover a rather broad range of GPCRs. Across the selected GPCRs, the split TEV assays for DRD1, DRD2, AVPR2, and GCGR performed best in HEK-293 cells as evidenced by the fold-change and Z′-factor values, both of which were the highest for many assays in this cell line. However, the HTR2A assays performed best in the PC12-TO cells, which was consistent with our previous report [16]. The GLP1R assays performed well in both the HEK-293 and PC12-TO cells.

The addition of SP (either alone or in combination with V2R) generally did not improve the split TEV assay performance (except marginally for DRD2). The added SP instead led to increased readout activity as observed in the absolute luciferase readings and also as supported by the Western blot analysis. We confirmed surface expression for native and a subset of SP-fused GPCRs linked to the NTEV moiety. Furthermore, GPCRs with a hemagglutinin-derived SP linked to full TEV moieties were previously shown to be efficiently transported to the cell surface [13]. We noted, however, that overexpressed GPCRs accumulated inside the cells. Thus, the SP may also have exacerbated an incorrect transport of overexpressed GPCRs in the cell, thereby precluding an improved assay performance. In some cases; i.e., for AVPR2 and GCGR in the HEK-293 cells, the addition of SP even reduced the fold changes strongly. Notably, GCGR contained an SP as part of its open reading frame [32]. Therefore, the endogenous SP provided the best assay performance, thereby making the addition of the additional SP obsolete [33]. Further, we point out that the activation of peptidergic GPCRs (in this study: AVPR2, GCGR, and GLP1R) could only occur at the cell surface because peptide ligands could not pass through the plasma membrane, supporting the notion of efficient surface expression.

The addition of the V2R tail improved the assay performance for DRD2 in the HEK-293, HeLa, and U-2 OS cells, and the addition of the SP further marginally improved the assay robustness as indicated by the Z′-factor values in the HEK-293 and U-2 OS cells. However, the other GPCR assays that were tested in the split TEV assays were not improved by the V2R tail. This contrasted with GPCR β-arrestin-2 recruitment assays that used a full TEV protease approach and were reported to be improved by the addition of V2R [13,17]. We also found that the DRD2 split TEV assays performed reasonably well in the HEK-293, HeLa, and U-2 OS cells even without a V2R tail, although less efficiently. DRD2 does not have serine and threonine residues in its C-terminal tail, which, when phosphorylated after receptor activation, are a common β-arrestin binding motif. However, β-arrestin-2 can also bind to phosphorylated serine and threonine motifs located within the third intracellular loop, which suggests that the V2R tail might be redundant or sensitive to the local environment [27,34]. In other cases, as for AVPR2, DRD1, and GCGR, the addition of the V2R tail did not improve the assay performance, but instead led to a substantial decrease in both the fold change and Z-factor. As summarized in Table 2, the native form followed by the V2R variant were the preferred variants to use depending on the GPCR. 

All of the split TEV assays were based on transient transfection with pcDNA3-based vectors, which are some of the most frequently used vectors in molecular and cellular biology [35]. However, we cannot rule out any differences in assay performance that may occur when using different vector backbones or different promoters (we used the CMV promoter; other options are, e.g., human EF1α or UbC promoters) for expressing split TEV fusion proteins. Furthermore, pcDNA3 expression plasmids harbor the bovine growth hormone polyadenylation signal sequence to increase expression in mammalian cells [36]. Therefore, we selected the pcDNA3 plasmid as the backbone, introduced corresponding GPCR cassettes via Gateway cloning, and validated the plasmid expression using Western blotting.

All of the assays presented here were based on transient transfections. We would like to note that rather low amounts of plasmids (c.f. the Section 2) were efficient in producing robust and sensitive assays. We regularly co-transfect a fluorescent marker plasmid (here: the nuclear form of EYFP) to assess efficiency. Therefore, we recommend always using clear plates for culturing the cells even if this requires a later transfer into white 96-well plates to measure the signals of the firefly luciferase. For the transfection reagent, we recommend testing with fluorescent marker plasmids first to find a suitable reagent because the performance can vary significantly from cell type to cell type. In addition, it may be helpful to co-transfect a constitutively active *Renilla* plasmid to assess the potential toxicity of compounds. 

For GCGR and GLP1R, the split TEV and pathway assays correlated well, including for the range of *EC_50_* values. However, the dose–response assays that used pathway sensors had a steeper Hill slope when compared to the split TEV assays, potentially implicating that the signals, which were initiated at the cell membrane and captured downstream, reached signal saturation via amplification inside the cell and/or were subject to feedback regulation [37,38].

GPCR biology is critical to complex disorders such as schizophrenia and type II diabetes. To provide alternative approaches for the discovery of therapeutic agents that simultaneously modulate DRD1, DRD2, and HTR2A or allosterically target GLP1R and the closely related GCGR, we developed and optimized genetically encoded cell-based split TEV assays for these targets to enable the profiling of compound libraries to identify selective agonists. DRD1, DRD2, and HTR2A have been implicated as drug targets in schizophrenia [39,40]. Schizophrenic patients suffering from positive symptoms such as hallucinations benefit from DRD2 and HTR2A antagonists [41], but potentially also from low-efficacy DRD1 agonists [42]. The anticipated medication paradigms are thus complex, and the development of drugs with a desired range of polypharmacology in disease-relevant targets is very demanding. Although each receptor can be efficiently targeted by many approved drugs (e.g., risperidone, which antagonizes DRD2 and HTR2A [43]; or lisuride, which is a partial agonist for DRD1 [44]), a multitarget drug that combines the features is still elusive. 

GCGR and GLP1R are key targets to treat type II diabetes [21,22]. For both GCGR and GLP1R, multiple ligands have been approved by the FDA and EMA [45,46,47], even as the orally administrable peptide analogon known as semaglutide [48,49]. Nevertheless, the development of small-molecule drugs with an improved polypharmacological profile and an increased half-life is desired, and the search as well as clinical testing are still ongoing [50,51,52]. Furthermore, the activation of GLP1R was implicated in neuroprotective effects in ALS [25], and GLP-1 mimetics could alleviate ALS relevant phenotypes in both cellular and murine models [53,54]. However, liraglutide was recently reported to have failed to exert beneficial effects in a mouse model of ALS [55], thus still requiring better drugs to be developed. 

Robust and sensitive assays using the technology outlined in this study could be applied in a multiplexed approach. In these assays, multiple experimental conditions can be monitored in parallel in a single well, and it is expected that such multiplexed assays will promote the broad screening for more selective drugs with fewer side activities [56,57]. In such multiparametric assays, each assay cell type expresses a different target linked to a unique barcode as the reporter and enables the pooling of single assays [16]. In addition, signaling cascades and pathways such as Gα_s_-dependent cAMP and Gα_q_-stimulated calcium responses or β-arrestin-2 mediated ERK1/2 signaling can be simultaneously monitored using barcoded CRE and EGR1 promotor (MAPK) pathway reporters, thus linking target and pathway responses. For example, each of the GPCR split TEV assays could be multiplexed with CRE and EGR1 promoter sensor pathway assays in one cell: different barcode reporters could be linked to the 10xUAS sensors used for split TEV assays, CRE sensors could be used to assess the cAMP/calcium signaling, and EGR1-promoter sensors could be used to assess the MAPK signaling. Such an approach may be particularly attractive to researchers in drug development because the monitoring of both GPCR activities and the elicited biased signaling actions are expected to generate better drugs with higher efficacies and reduced adverse effects [10,58].

## 5. Conclusions

Taken together, we provided a technical resource for optimizing split TEV GPCR β-arrestin-2 recruitment assays. For each unique GPCR, the effects of genetic modifications; i.e., the addition of an SP and/or a V2R tail, and the cell line of choice should be thoroughly tested to determine the best assay performance.

## Figures and Tables

**Figure 1 biosensors-13-00048-f001:**
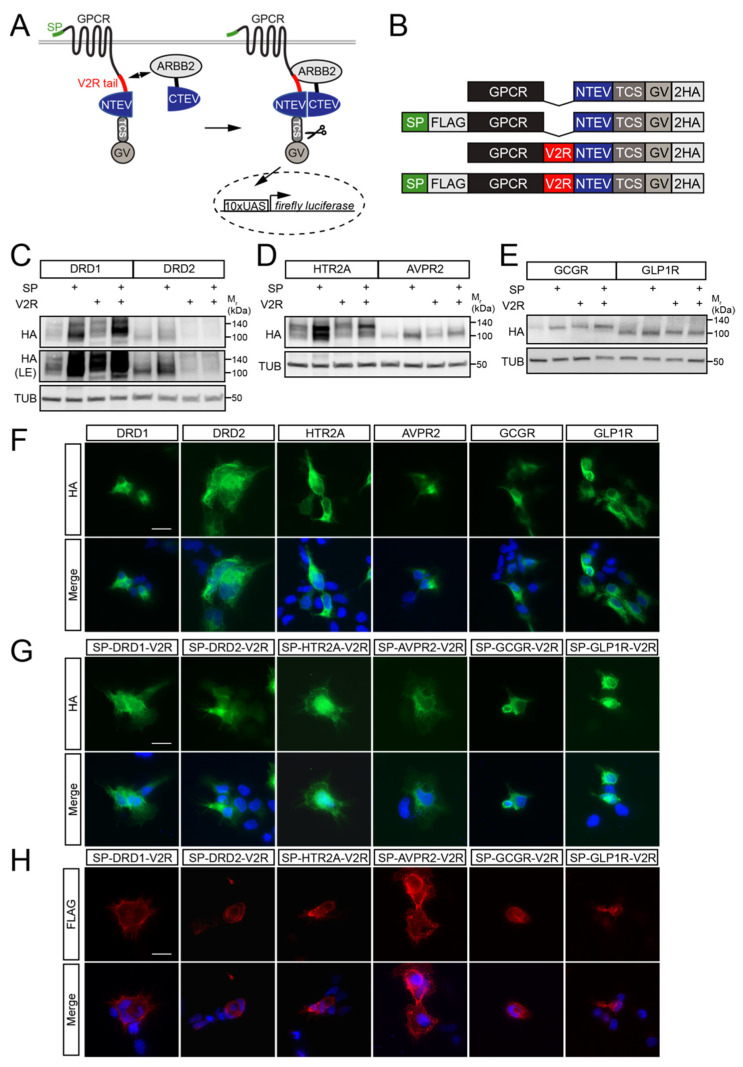
GPCRs for split TEV β-arrestin-2 recruitment assays were modified by the addition of a signal peptide, a V2R tail, or both. (**A**) Scheme of the GPCR β-arrestin-2 split TEV recruitment assay. A GPCR could be fused to the signal peptide (SP) and/or the tail of vasopressin receptor 2 (V2R) to test assay performance. ARRB2, β-arrestin-2; TCS, TEV protease cleavage site. (**B**) Graphical representation of the split TEV GPCR fusions. Depicted are the four variants of the GPCRs tested: native GPCR, SP-GPCR fusion, GPCR-V2R fusion, and SP-GPCR-V2R fusion. FLAG, single FLAG tag; NTEV, N-terminal moiety of the TEV protease; TCS, TEV protease cleavage site; GV, synthetic co-transcriptional activator GAL4-VP16; 2HA, double HA tag. (**C**–**E**) Split TEV GPCR constructs were properly expressed in HEK-293 cells. Western blots of the split TEV GPCR fusion constructs for DRD1 and DRD2 (**C**), HTR2A and AVPR2 (**D**), and GCGR and GLP1R (**E**). All proteins were detected with an HA antibody. LE, longer exposure. (**F**–**H**) Surface expression of selected GPCR split TEV fusions using immunocytochemistry staining. Native GPCR split TEV fusions were stained against the cytosolic HA epitope (**F**). SP-GPCR-V2R split TEV fusions were stained against the cytosolic HA epitope (**G**) and an extracellular FLAG epitope that was located C-terminal to SP (**H**). Note that FLAG staining was performed without permeabilizing cells. The merge was Dapi and antibody staining. Scale bar = 20 µM.

**Figure 2 biosensors-13-00048-f002:**
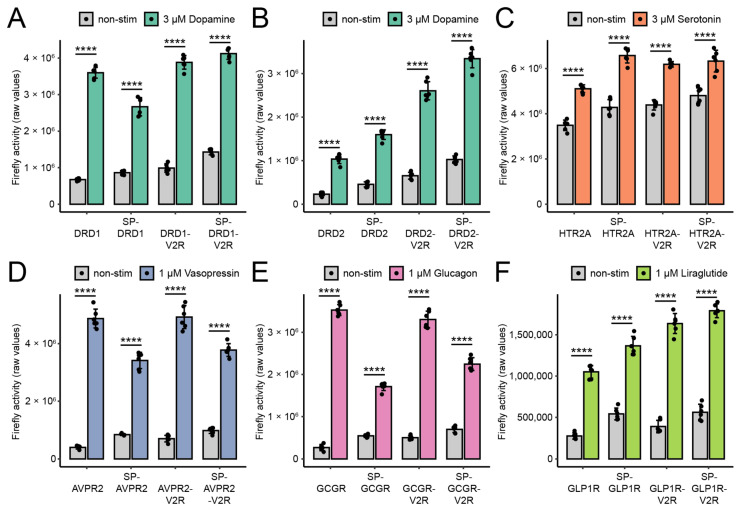
Split TEV GPCR β-arrestin-2 recruitment assays in HEK-293 cells. (**A**–**F**) Luciferase end-point assays for DRD1 (**A**), DRD2 (**B**), HTR2A (**C**), AVPR2 (**D**), GCGR (**E**), and GLP1R (**F**). All assays were conducted in HEK-293 cells in a 96-well format. Cells were stimulated for 6 h with their agonists. Note that the addition of the signal peptide and the V2R tail affected the performance of the split TEV GPCR β-arrestin-2 recruitment assays. Bar graphs display the means; the error bars represent the s.d. with six replicates per conditions. A two-tailed Student’s t test was used to determine the *p*-values for treatment versus control. **** *p* ≤ 0.0001.

**Figure 3 biosensors-13-00048-f003:**
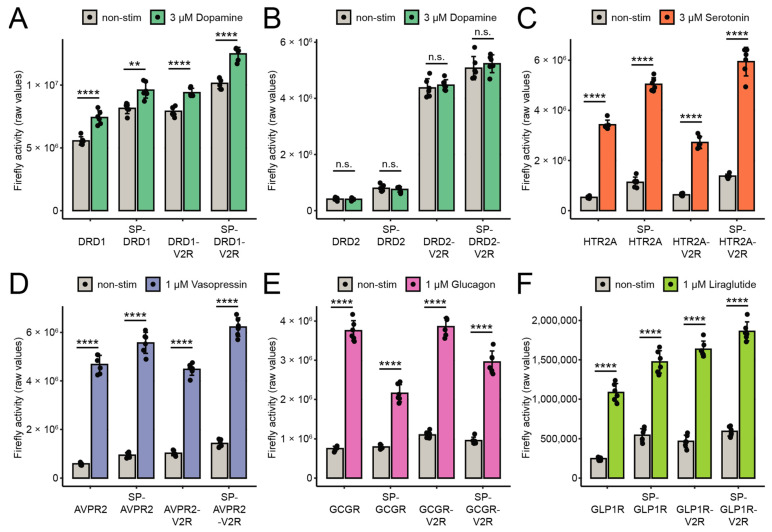
Split TEV GPCR β-arrestin-2 recruitment assays in PC12-TO cells. (**A**–**F**) Luciferase end-point assays for DRD1 (**A**), DRD2 (**B**), HTR2A (**C**), AVPR2 (**D**), GCGR (**E**), and GLP1R (**F**). All assays were conducted in a 96-well format. Assays were stimulated for 6 h with their agonists. Bar graphs display the means; the error bars represent the s.d. with six replicates per conditions. A two-tailed Student’s *t* test was used to determine the *p*-values for treatment versus control. ** *p* ≤ 0.01; **** *p* ≤ 0.0001, n. s., not significant.

**Figure 4 biosensors-13-00048-f004:**
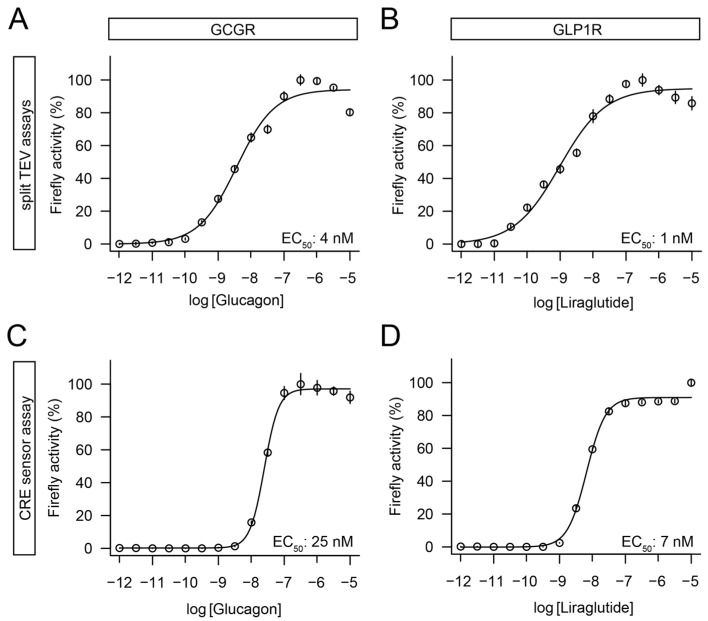
Dose–response curve analysis identified robust agonist assays for GCGR and GLP1R using both target-based split TEV and pathway-based CRE reporters. (**A**) Dose–response assay for native GCGR using glucagon as stimulus and split TEV as readout. Hill slope = 0.741. (**B**) Dose–response assay for GLP1R-V2R using liraglutide as stimulus and split TEV as readout. Hill slope = 0.602. (**C**) Dose–response assay for native GCGR split TEV construct as used in (**A**) using glucagon as stimulus and a CRE reporter as readout. Hill slope = 1.991. (**D**) Dose–response assay for GLP1R-V2R split TEV construct as used in (**B**) using liraglutide as stimulus and a CRE reporter as readout. Hill slope = 1.491. Receptors were stimulated for 6 h. Data represent mean ± s.e.m. Firefly counts were normalized to the minimum and maximum of ligand response.

**Table 1 biosensors-13-00048-t001:** Robustness of split TEV GPCR assays. Fold-change and Z′-factor values of all split TEV GPCR β-arrestin-2 assays that were conducted in HEK-293, HeLa, U 2-OS, and PC12-TO cells.

HEK-293	DRD1		DRD2		HTR2A		AVPR2		GCGR		GLP1R	
	FC	Z′-Factor	FC	Z′-Factor	FC	Z′-Factor	FC	Z′-Factor	FC	Z′-Factor	FC	Z′-Factor
Native	**5.34 ^1^**	**0.81**	**4.52**	0.46	1.46	0.25	**12.37**	**0.74**	**13.32**	**0.81**	3.82	**0.56**
SP	3.09	0.50	3.51	0.55	**1.54**	0.12	4.07	0.63	3.15	0.67	2.52	0.32
V2R tail	3.92	0.69	3.99	0.56	1.41	**0.42**	7.06	0.63	6.63	0.74	**4.20**	0.53
SP and V2R tail	2.89	0.74	3.26	**0.62**	1.32	−0.63	3.85	0.65	3.22	0.56	3.19	0.56
**HeLa**	DRD1		DRD2		HTR2A		AVPR2		GCGR		GLP1R	
	FC	Z′-factor	FC	Z′-factor	FC	Z′-factor	FC	Z′-factor	FC	Z′-factor	FC	Z′-factor
Native	**2.33**	0.28	**3.46**	0.09	**1.41**	**0.17**	1.42	−1.93	1.31	−0.39	**1.43**	**−0.48**
SP	1.64	−0.27	2.05	−0.98	(1.03) ^2^	(−7.90)	**2.54**	0.05	1.14	−2.90	1.26	−0.97
V2R tail	2.18	0.24	2.93	**0.57**	1.15	−2.40	1.52	−1.04	**1.56**	**0.25**	(1.06)	(−6.53)
SP and V2R tail	2.27	**0.44**	2.99	0.46	(0.97)	(−14.80)	2.08	**0.44**	0.75	−0.46	(1.10)	(−4.71)
**U-2 OS**	DRD1		DRD2		HTR2A		AVPR2		GCGR		GLP1R	
	FC	Z′-factor	FC	Z′-factor	FC	Z′-factor	FC	Z′-factor	FC	Z′-factor	FC	Z′-factor
Native	**2.06**	0.38	1.41	0.30	**1.12**	**−2.56**	**1.86**	**0.36**	**1.47**	−0.29	1.20	−1.99
SP	1.83	**0.45**	1.79	0.19	(1.07)	(−5.23)	1.61	0.13	1.18	−1.45	**1.40**	−0.55
V2R tail	1.72	0.32	**2.71**	0.65	(1.06)	(−7.04)	1.67	0.16	1.46	**0.07**	1.36	**−0.11**
SP and V2R tail	1.84	0.37	2.67	**0.69**	(1.07)	(−4.35)	1.57	0.31	1.28	−0.40	1.29	−0.52
**PC12-TO**	DRD1		DRD2		HTR2A		AVPR2		GCGR		GLP1R	
	FC	Z′-factor	FC	Z′-factor	FC	Z′-factor	FC	Z′-factor	FC	Z′-factor	FC	Z′-factor
Native	**1.33**	−0.40	(0.98)	(−38.38)	**6.45**	**0.76**	**7.92**	0.68	**4.99**	**0.68**	**4.38**	0.53
SP	1.18	−1.18	(0.95)	(−14.82)	4.45	0.64	5.89	0.65	2.72	0.40	2.71	0.28
V2R tail	1.19	−0.46	(1.02)	(−15.37)	4.29	0.58	4.36	**0.69**	3.50	0.66	3.49	0.53
SP and V2R tail	1.23	**−0.21**	**(1.03)**	**(−13.06)**	4.32	0.57	4.35	0.66	3.09	0.45	3.15	**0.58**

^1^ Numbers highlighted in bold indicate the highest fold-change and Z′-factor values for each GPCR in each cell line; i.e., considering the four variants of each GPCR used (native form, SP-GPCR variant, GPCR-V2R variant, and SP-GPCR-V2R variant). Results were calculated from experiments with 6 replicates. ^2^ Numbers in parentheses indicate that the *p*-value was larger than 0.05 as calculated via the two-sided Student’s *t* test.

**Table 2 biosensors-13-00048-t002:** Preferred GPCR variant and cell line for split TEV assays: effects of signal peptide (SP), V2R tail, and cell line on assay performance. The best variant was identified by the highest fold change and a paralleled Z′-factor ≥ 0.5 across the cell lines tested.

Target	Preferred Variant	Preferred Cell Line
DRD1	Native	HEK-293
DRD2	V2R variant ^1^	HEK-293
HTR2A	Native	PC12-TO
AVPR2	Native	HEK-293
GCGR	Native	HEK-293
GLP1R	Native ^2^, V2R variant ^3^	PC12-TO ^2^, HEK-293 ^3^

^1^ The DRD2-V2R variant had the highest fold change for a Z′-factor ≥ 0.5. ^2^ The fold change for GLP1R was highest in PC12-TO cells with the native version. ^3^ The use of the V2R variant in HEK-293 cells was preferred due to compatibility with other GPCR assays.

## Data Availability

Raw data of split TEV GPCR assays are available at Mendeley Data. DOI: 10.17632/fd45dm2bz5.1.

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
