# Peer review of "Improved Split TEV GPCR β-arrestin-2 Recruitment Assays via Systematic Analysis of Signal Peptide and β-arrestin Binding Motif Variants"

_biosensors, 2022, doi:10.3390/bios13010048_

Round 1
Reviewer 1 Report
This study evaluates the addition of a signal sequence peptide and a V2 vasopressin receptor sequence to the function of split TEV GPCR reporters. The authors build on previous work with four class A receptors and add two new class B GPCRs using this split TEV reporter mechanism. Additionally, the study examines the cell line differences for the various reporter construct performance. While the native GPCR proves to typically be the appropriate choice for the assay, troublesome GPCRs may benefit from the noted constructs outlined herein.
Overall the work is clear in its expansion of the split TEV tags and would be useful for those building future assays off of this technology. The results are presented in a very comprehensible manner. My one concern is the minimal information on plasmid and transfection reagent concentration for each cell type. Transfection efficiency can vary depending on cell type and may require optimization. It’s also possible that different concentrations of the GPCR DNA construct may result in better assay performance. At a minimum I would be interested to have the transfection amounts be added to the methods section as well as a discussion of any helpful transfection troubleshooting for the reader’s interest.
It is likely outside the scope of this study, but I think it would be beneficial to know that the various constructs are indeed expressed on the cell surface, possibly using ELISA or FACS. One would expect the signal peptide to help with transport to the cell surface, but perhaps incorrect transport is why the higher expression does not yield an improved assay.
Questions I have while reading the text:
- Is there a hypothesis as to why the different cell lines had different reporter success?
- Could other signal peptides or c-terminal additions be used instead of the ones shown here?
Comments on specific figures and lines of text:
Fig3: Is there a hypothesis for high non-stim activation of DRD1 and DRD2 in PC12-TO cells?
Fig4: What is the error on dose response EC50s? Are then two assays actually significantly different?
Relatedly:
Line 360- should this language be so sure that these are distinct?
Line 367- can you report the hill slope somewhere if you’re going to talk about it?
Paragraphs starting in lines 370 and 379 feel a bit out of the blue. I think they would benefit if you tied in your assay technologies at the start, for example moving the sentence in line 387 to the start to motivate these two big picture paragraphs.
Line 392- I think this paragraph could benefit from a more explicit tie-in with the assay developments outlined in the paper. “Robust and sensitive assays using the technology outlined in this study could be applied in a multiplexed approach…”
Reviewer 2 Report
The purpose of this study is to improve the performance of the split TEV-based GPCR-mediated beta-arrestin2 recruitment assay. GPCRs are therapeutically relevant drug targets, and such an assay may be useful for drug discovery purposes, but improvements are needed. Beta-arrestins are typically recruited to agonist-activated and phosphorylated GPCRs, and the split TEV monitors the interaction of an agonist-activated GPCR with beta-arrestin2 through TEV protein fragment complementation using a luminescent signal as readout. Here the authors attempted to optimize the recruitment assay using six highly relevant GPCRs (DRD1, DRD2, HTR2A, GCGR, AVPR2, and GLP1R). Four variants of each GPCR was examined across four cell lines (HEK-293, PC12 Tet-Off, U-2 OS, and HeLa). Overall this is a timely study and the results are interesting, but unfortunately, no general consensus emerged as to the optimal assay conditions as assay performance differed significantly for each GPCR variant and seemed to be dependent on the cell line. However, the findings will be of interest to the field as researchers attempt to optimize this particular beta-arrestin recruitment assay for their favorite GPCR drug discovery program. I have some minor concerns that should be addressed by the authors.
1. The expression of each receptor variant was examined by Western blotting, which does not provide important information about the surface-level expression of the receptors. The authors should acknowledge this limitation in the discussion, although it would strengthen the conclusions if surface expression experiments were performed for each variant. Such experiments/data may help to better explain the performance of the variants or cell-line dependent differences.
2. Unfortunately, I was unable to access the supplemental materials, although most of the relevant information seems to be included in Table 1 the data within the supplemental materials should be included.
3. While the manuscript is mostly well-written and the data are logically presented, some passages were difficult to understand. The authors should revise sentences in lines 58 through 60 and 77 through 79.
Round 2
Reviewer 2 Report
The authors have done a nice job addressing my concerns and comments. They provided new data showing surface expression of GPCRs and improved the discussion. Congratulations on a nice study.